Manuscript prepared for Atmos. Chem. Phys.
with version 2015/04/24 7.83 Copernicus papers of the LaTeX class copernicus.cls.
Date: 31 October 2016

# First detection of ammonia ($NH_3$) in the Asian summer monsoon upper troposphere

Michael Höpfner[1], Rainer Volkamer[2,3], Udo Grabowski[1], Michel Grutter[4], Johannes Orphal[1], Gabriele Stiller[1], Thomas von Clarmann[1], and Gerald Wetzel[1]

[1]Institute of Meteorology and Climate Research, Karlsruhe Institute of Technology, Karlsruhe, Germany.
[2]Department of Chemistry & Biochemistry, University of Colorado, Boulder, CO, USA.
[3]Cooperative Institute for Research in Environmental Sciences, University of Colorado at Boulder, CO, USA.
[4]Centro de Ciencias de la Atmósfera, Universidad Nacional Autónoma de México, Mexico City, Mexico.

*Correspondence to:* M. Höpfner (michael.hoepfner@kit.edu)

**Abstract.** Ammonia ($NH_3$) has been detected in the upper troposphere by analysis of averaged MIPAS (Michelson Interferomter for Passive Atmospheric Sounding) infrared limb-emission spectra. We have found enhanced amounts of $NH_3$ within the region of the Asian summer monsoon at 12–15 km altitude. Three-monthly, $10°$ longitude $\times$ $10°$ latitude average profiles reaching maximum
mixing ratios of around 30 pptv in this altitude range have been retrieved with a vertical resolution of 3–8 km and estimated errors of about 5 pptv. These observations show that loss processes during transport from the boundary layer to the upper troposphere within the Asian monsoon do not deplete the air entirely of $NH_3$. Thus, ammonia might contribute to the so-called Asian tropopause aerosol layer by formation of ammonium aerosol particles. On a global scale, outside the monsoon area and
during different seasons, we could not detect enhanced values of $NH_3$ above the actual detection limit of about 3-5 pptv. This upper bound helps to constrain global model simulations.

## 1  Introduction

In the Earth's atmosphere the trace gas ammonia ($NH_3$) represents the major form of reduced nitrogen. With a share of around 70–80%, the bulk of ammonia emissions is due to anthropogenic
activity, namely the use of synthetic fertilizers and livestock manure management (Bouwman et al., 1997; Paulot et al., 2015). Major source regions of $NH_3$ are located in south-east China and northern India (Paulot et al., 2014; Van Damme et al., 2015).

Neutralization of acids by the alkaline gas $NH_3$ leads to the formation of ammonium salts in the atmosphere. For example, reaction of $NH_3$ with sulfuric acid ($H_2SO_4$) or nitric acid ($HNO_3$) forms
aerosol particles composed of ammonium sulfate, $(NH_4)_2SO_4$ , or ammonium nitrate, $NH_4NO_3$ (e.g. Behera et al., 2013, and references therein). These inorganic aerosols are important not only

with regard to air quality considerations (Hamaoui-Laguel et al., 2014) but they also affect climate through various direct and indirect radiative impacts (Adams et al., 2001; Martin et al., 2004; Liao and Seinfeld, 2005; Forster et al., 2007; Xu and Penner, 2012; Boucher et al., 2013). Further, cirrus clouds might also be affected by the presence of $NH_3$ and ammonium (Tabazadeh and Toon, 1998; Wang et al., 2008). E.g. ammonium sulfate aerosols that are partially coated and have exposed surface sites are active with respect to ice nucleation (Prenni et al., 2001; Wise et al., 2004). Such a heterogeneous nucleation pathway might influence size and number of cirrus particles and, consequently their radiative impact (Abbatt et al., 2006). Moreover, through stabilization of sulfuric acid clusters, ammonia itself may play an important role regarding the initial nucleation of sulfuric acid aerosols (Ortega et al., 2008; Kirkby et al., 2011; Schobesberger et al., 2013; Kürten et al., 2015).

Global emissions of $NH_3$ are expected to rise strongly due to the need to sustain a growing population and due to enhanced emissions under increasing temperatures (Erisman et al., 2008; Vuuren et al., 2011; Sutton et al., 2013). As a result, in future prospects for a positive radiative forcing by a decrease of the shortwave albedo caused by reductions of industrial $SO_2$ emissions may partly be compensated by increasing amounts of ammonium containing aerosols (Bellouin et al., 2011; Xu and Penner, 2012; Shindell et al., 2013; Hauglustaine et al., 2014).

With regard to the predicted increase in $NH_3$ emissions and the possible compensating effect on aerosol radiative forcing, Paulot et al. (2016) emphasize the need to better constrain also the vertical distribution of ammonia. However, there is very little information from measurements on $NH_3$ at mid- and upper tropospheric levels.

Before 2008, measurements of ammonia were almost exclusively based on in-situ technologies (von Bobrutzki et al., 2010). Compared to the wealth of observations on ground, vertical profiles of $NH_3$ from in-situ observations above the boundary layer are relatively sparse. Recently, aircraft-borne campaign measurements over the US obtained concentration profiles in the free troposphere reaching altitudes of about 6 km (Nowak et al., 2007; Nowak et al., 2010, 2012; Leen et al., 2013; Schiferl et al., 2016). At these altitudes maximum observed $NH_3$ mixing ratios reached about 800 pptv (Schiferl et al., 2016) with detection limits of 70 pptv (Nowak et al., 2010). In-situ air-borne observations over Germany by Ziereis and Arnold (1986) restricted concentrations to the sub-pptv range at altitudes between 8 and 10 km . To our knowledge, these are the only upper tropospheric in-situ-measurements of $NH_3$ published so far.

A first step in the direction of observations with global coverage was achieved by Beer et al. (2008), who reported the detection of $NH_3$ in the lower troposphere from space-borne nadir sounding measurements by the Tropospheric Emission Spectrometer (TES) on the EOS Aura satellite. Subsequently, various papers have been published describing retrieval, validation and interpretation of $NH_3$ derived from the nadir sounders TES (Clarisse et al., 2010; Shephard et al., 2011), IASI (Infrared Atmospheric Sounding Interferometer) (Coheur et al., 2009; Clarisse et al., 2009, 2010; Van Damme et al., 2014), CrIS (Cross-track Infrared Sounder) (Shephard and Cady-Pereira, 2015),

and AIRS (Atmospheric Infrared Sounder) (Warner et al., 2016). The vertical sensitivity of these satellite retrievals is mainly limited to the lower troposphere up to about 3–4 km and no altitude resolution is achieved (e.g., Clarisse et al., 2010; Shephard and Cady-Pereira, 2015). Recently, retrievals of $NH_3$ vertical column amounts from ground-based FTIR solar observations located at various sites have been presented by Dammers et al. (2015) and are being used for the quantitative validation of space-borne nadir-viewing datasets (Dammers et al., 2016). As shown by Dammers et al. (2015) in case of high amounts of $NH_3$ near the surface, the retrieval sensitivity peaks within the boundary layer where also the altitude-gradient can be derived. For low concentrations, the retrieval is only sensitive to the total vertical column amount.

To achieve vertically resolved profiles of trace gases in the upper troposphere and above, limb-sounding techniques have been applied frequently. Regarding ammonia, Oelhaf et al. (1983) reported upper limits of 100 pptv above 10 km by analysis of balloon-borne limb solar absorption spectra measured over the US. Space-borne solar occultation measurements obtained with the ACE-FTS instrument within a plume of biomass burning over Tanzania have been studied by Coheur et al. (2007). In that publication, the authors mention a "tentative identification" of $NH_3$ in the spectra, while the spectral signals of various other trace species, such as $C_2H_4$, $C_3H_6O$, $H_2CO$ and PAN were detected unequivocally. Nonetheless, Coheur et al. (2007) derived a vertical profile of $NH_3$ between 6.5 and 17 km with typical values of less than 20 pptv and a maximum of about 50 pptv at 8 km. Burgess et al. (2006) report on first attempts to retrieve global distributions of ammonia using limb infrared emission spectra measured by the Michelson Interferometer for Passive Atmospheric Sounding (MIPAS) on Envisat. They obtained one climatological vertical profile with $NH_3$ volume mixing ratios below 5 pptv at altitudes above 9 km. However, no evidence for the presence of $NH_3$ in the limb spectra nor any indication for enhanced values within the area of the Asian summer monsoon is shown.

In summary, considering the reported observations, neither the in-situ measurements by Ziereis and Arnold (1986) nor the limb-sounding remote sensing data (Oelhaf et al., 1983; Coheur et al., 2007; Burgess et al., 2006) prove the presence of ammonia at altitudes above about 8 km. In the work presented below we show, to our knowledge, the first evidence for $NH_3$ together with quantitative retrievals at upper tropospheric levels by use of MIPAS averaged limb spectra.

## 2 MIPAS/Envisat

On board the Envisat satellite the MIPAS limb sounder recorded infrared spectra of the radiation emitted by atmospheric constituents from June 2002 until April 2012 (Fischer et al., 2008). Between June 2002 and March 2004 (period 1), the spectral resolution was $0.025\,cm^{-1}$ with one limb-scan consisting of 17 spectra from about 6–60 km altitude with steps of 3 km up to about 42 km in the case of nominal mode observations. From January 2005 until April 2012 (period 2), the spectral

resolution was degraded to $0.0625\,\mathrm{cm}^{-1}$. This was accompanied by a better vertical sampling (27 tangent levels up to about 70 km altitude with 1.5 km steps up to $\approx$23 km). Also in the horizontal direction along the satellite track the sampling improved from a distance between subsequent limb-scans of 550 km during period 1 to 420 km during period 2.

## 3 Retrieval and spectral detection

Here we report on the detection and retrieval of $NH_3$ from MIPAS observations in the upper troposphere on the basis of averaged limb-spectra. This method has already been applied successfully for the detection of bromine nitrate ($BrONO_2$) (Höpfner et al., 2009) and for the compilation of a global climatology of stratospheric sulfur dioxide ($SO_2$) from MIPAS measurements (Höpfner et al., 2013). In those investigations the mean spectra consisted of monthly zonal averages within $10°$ latitude intervals, whereas for the present work we have chosen seasonal (3-monthly) averages within bins of $10°$ latitude by $10°$ longitude. Thus, we have refrained from zonal averaging in order to obtain resolution in the meridional direction, albeit slightly sacrificing temporal resolution. To reduce the spectral noise by at least a factor of five, we have chosen a lower limit of 25 single spectra (MIPAS level-1b version 5) for averaging. The resulting mean number of co-added spectra per time/latitude/longitude bin varies from 53–56 for the years 2003 and 2007 to 65–70 for 2008–2011 reaching maximum numbers of around 140 (see supplemental material for the detailed geographical and temporal distribution). This leads to a typical reduction of the spectral noise by 0.1 ranging from 0.2 to 0.08 and signal-to-noise values resulting in retrieval errors near and below 1 pptv of $NH_3$ (see detailed error estimation below). In the troposphere the number of available spectra is limited as a result of cloud contamination along the limb line-of-sight. We have applied a cloud filter to de-select cloud-contaminated spectra before averaging. For the cloud detection scheme the established cloud-index method (Spang et al., 2004) with a cloud index limit of 2.0 has been used.

To derive altitude profiles of $NH_3$ from each averaged limb-scan we have applied a constrained non-linear multi-parameter least-squares fitting procedure whereby measurements from all spectra of one limb scan are analysed in one step (e.g., von Clarmann et al., 2003; Höpfner et al., 2009). The unknown atmospheric state is described in terms of trace gas volume mixing ratios at discrete altitude levels with a grid distance of 1 km. This grid is finer compared to the instrumental vertical field-of-view width of about 3 km at the tangent points and also finer than the vertical sampling distance of 1.5–3 km. To dampen vertical oscillations arising from the ill-posedness of the inverse problem a first order smoothing constraint has been chosen (Tikhonov, 1963; Steck, 2002). The regularization strengths have been adjusted independently for each species being retrieved simultaneously.

For fitting of the $NH_3$ signatures we have chosen spectral windows within the interval 950–$970\,\mathrm{cm}^{-1}$, which have the advantage that they are situated in the region of one of the optically thinnest mid-infrared atmospheric windows. Furthermore, there are relatively few spectrally inter-

fering species which have to be retrieved simultaneously with $NH_3$. Spectroscopic line parameters
of the HITRAN 2012 database (Rothman et al., 2013) have been used.

A scheme consisting of two steps has been identified as adequate for the retrieval of $NH_3$. First,
the broader wavenumber range 962–968 $cm^{-1}$ has been chosen to fit the strong $CO_2$ lines of the
laser band together with the interfering species $O_3, H_2O, NH_3, COF_2$. In the second step, narrow
analysis windows have been placed around the strongest signatures of $NH_3$: 951.6–952.0 $cm^{-1}$,
965.1–965.6 $cm^{-1}$, and 966.6–967.5 $cm^{-1}$ (MIPAS period 1) and 951.625–952.0 $cm^{-1}$, 965.125–
965.625 $cm^{-1}$, and 966.625–967.5 $cm^{-1}$ (MIPAS period 2), thereby avoiding the peaks of the strong
$CO_2$ lines. At this stage $CO_2$ is kept fixed to the results from the initial retrieval, while $O_3, H_2O$ and
$COF_2$ are retrieved jointly with $NH_3$.

Figure 1 presents the spectral fit and the detection of $NH_3$ for examples from both MIPAS periods
at tangent heights around 12.5 km within the region of the Asian monsoon in June-August 2003 (top
panel) and 2009 (bottom panel). Within each panel the top row shows the observations in black, the
fit without taking $NH_3$ into account in blue and the retrieval including $NH_3$ in red. In the second
row of each panel the residual spectra are shown for the retrieval without (blue) and with (red)
consideration of $NH_3$. Here the green line is the difference between the two simulations (with minus
without $NH_3$) in order to show the spectral signature of $NH_3$ without any instrumental effect, such
as spectral noise.

The top panel of Fig. 1 reveals clearly the presence of $NH_3$ in MIPAS limb spectra. Radiative
transfer simulations without consideration of $NH_3$ lead to largest residuals (bold blue curves) at the
position of the ammonia lines (dotted orange curves). Only when ammonia is taken into account are
the observed spectra within all three microwindows fitted sufficiently well. Comparing the first row
of the bottom panel in Fig. 1 with that of the top panel, the worse spectral resolution of MIPAS
period 2 becomes obvious. Still the residuals of the $NH_3$ spectral lines and the better fit upon their
consideration are visible, especially in microwindows 2 and 3.

Results of the altitude dependent error estimation are presented in Fig. 2 for the two examples
of the limb scans for which the spectral fits have been shown in Fig. 1. A summary of the assumptions on the various sources of uncertainty is provided in Table 1. For spectral noise, the actual error
numbers referring to the two limb-scans discussed are given together with their range over all observations in brackets. While noise is directly mapped into the state space for each individual retrieval,
the error estimation for all other uncertainties has been performed by sensitivity calculations for
atmospheric conditions representative for observations within the influence of the Asian monsoon
(Höpfner et al., 2009; Höpfner et al., 2013).

In the left panels of Fig. 2 the bold dotted curves indicate the reconstructed vertical profiles
of $NH_3$. The concentrations reach maximum values of around 24–29 pptv. The bold solid lines
represent the total error calculated as the square root of the sum of all squared error components.
The total errors amount to around 2–6 pptv (17–80%, right panels) in the altitude region up to about

20 km. Above, the estimated errors are larger than the mixing ratios of $NH_3$. The leading error components are tangent altitude uncertainties, uncertainties in the HITRAN line intensity data of $NH_3$ and nonlinearity effects in the averaging procedure as discussed in Höpfner et al. (2009). On the other hand, the use of averaged spectra reduces the noise term to less than 1 pptv within the altitude range of interest.

The vertical resolution of the resulting altitude profiles of $NH_3$ mixing ratios is directly connected to the noise error values through the applied setting of the regularization strength. The altitude resolution of the retrieval is described by the retrieval averaging kernel matrix (Rodgers, 2000). Examples for both MIPAS periods are provided in Fig. 3. From these, typical vertical resolutions are derived as the retrieval grid width divided by the inverse of the diagonal matrix elements (Rodgers, 2000). The globally average vertical resolution at the altitude levels 12, 15, and 18 km, which are discussed in more detail below, is 6.6 km, 7.9 km and 8.8 km during period 1 and 3.5 km, 4.3 km, and 5.6 km during period 2.

## 4 The global dataset

Retrievals of $NH_3$ have been performed for the entire period of MIPAS observations, i.e. from July 2002 until April 2012. Figure 4 presents the global volume mixing ratio distributions at 15 km altitude during seven seasons from July 2002 until February 2004. There are enhanced values of up to 33 pptv within a region between 30°–110°E and 20°–50°N during boreal summer, coinciding with the occurrence of the Asian monsoons. During all other seasons of the two MIPAS periods and outside the region influenced by the Asian monsoon, no similarly high concentrations of $NH_3$ can be found within the entire altitude region covered by our measurements.

An overview for all years with sufficient data coverage in the Asian summer monsoon season during the MIPAS mission lifetime is provided in Fig. 5 for altitude levels of 12, 15, and 18 km. Due to the less frequent presence of clouds, the number of pixels with valid measurements increases with altitude. Similar to the results from MIPAS period 1, also during period 2 the enhancement of $NH_3$ within the Asian monsoon region is present for all years of observation. Further, on a global scale there are no other areas visible in the dataset with similarly enhanced values of $NH_3$. While at 12 and 15 km altitude $NH_3$ enhancements compared to the global background state are visible during all years, at 18 km altitude increased values of $NH_3$ are present only during the years 2003, 2008, and 2010.

A comparison between vertical profiles of $NH_3$ averaged within the western (30–70°E, solid lines) and eastern (70–110°E, dashed lines) parts of the monsoon region for the latitude band 30–40°N is presented in Fig. 6 for the years 2003 and 2007–2011. In the same Figure, the dotted curves show the $NH_3$ mean Jun/Jul/Aug profiles for all years outside the Asian monsoon area, for the same longitude and latitude range (30–110°E, 30–40°S) of the southern hemisphere. The profiles in the region of

the Asian monsoon reveal that the maximum concentrations over the whole altitude range within one year are always larger in the eastern part of the Asian monsoon compared to the western part. Maximum concentrations of $NH_3$ in the eastern part reach about 10–22 pptv at 11-13 km altitude. Largest values are found in 2003 and 2009 and lowest ones in 2007 and 2011.

In the western part of the area influenced by the Asian monsoon, enhanced averaged volume mixing ratios of $NH_3$ can be observed during the years 2003, 2008, and 2010 with values ranging from 6 to 15 pptv. Situated at around 13–15 km, the maximum concentrations in the western part are always located at higher altitudes compared to those from the eastern part of the monsoon region. The position of the $NH_3$ maximum at higher altitudes in the western compared to the eastern part of the monsoon system might be due to convective uplift of boundary layer air in the east followed by upper tropospheric transport and further uplift towards the west. Such an uplift of air from east to west is indicated in Vogel et al. (2014, Fig. 10) by trajectory calculations, however mainly located at the border of the anticyclone.

The mean $NH_3$ profiles of the western part show no clear enhancements during the years 2007, 2009 and 2011. These profiles exhibit maximum values below 5 pptv, which are in the range of concentrations retrieved in the case of the 'background' state of the southern hemisphere (indicated by dotted lines in Fig. 6). These values are below our estimated detection limit (see below).

Due to the lack of ammonia observations in the upper troposphere, we cannot validate our dataset with correlative measurements. However, in the next section we discuss its plausibility by comparing with the few previous observations and atmospheric model results.

## 5 Discussion

As mentioned in the introduction, observations of $NH_3$ reaching upper tropospheric levels have been published by Ziereis and Arnold (1986). They report upper limits of about 0.04 pptv between 8 and 10 km over Germany in May 1985. At the present state of our MIPAS data analysis we cannot contradict those upper values outside the influence of the Asian summer monsoon system. Given the total error of a few pptv we would estimate the 1-$\sigma$ detection limit of our retrieval to be about 3–5 pptv. One might argue that the use of a 1-$\sigma$ detection limit does not provide sufficiently significant evidence of the $NH_3$ enhancement within the monsoon. However, random errors cannot explain why the enhancements should appear in a contiguous geographical pattern nor could they account for the enhancements appearing only in one season.

For example within the data shown in Fig. 5 at 12 km, there are 176 values larger than 5 pptv outside the region 20–50°N × 30-120°E compared to 55 inside. However, at the 15 km level, only 5 data points exceed 5 pptv outside but 37 inside. Using 2-$\sigma$, there are no data points outside compared to 23 and 15 exceeding 10 pptv inside the region at 12 km and 15 km, respectively. Further, the detected enhancements inside the monsoon region are in many cases (13 times at 12 km and 8 times

at 15 km) even above 15 pptv and, thus, larger than a 3-$\sigma$ limit. Temporally resolved, values above 10 pptv in the monsoon region exist during all years at both altitude levels with the exception of 2011 at 15 km. 15 pptv are exceeded at 10 km in 2003–2010 and at 15 km in all years but 2007 and 2011.

Regarding the retrievals outside the monsoon area, there is a difference between the two MIPAS measurement periods at 10–12 km altitude (see e.g. the difference in the dotted lines in Fig. 6 or the higher background level at 12 km altitude visible in Fig. 5 between the year 2003 and 2007–2011) that reaches 4 pptv. We attribute this discrepancy to an unexplained systematic uncertainty caused by the different spectral resolutions between the two instrumental states. This observation

corroborates our error estimation and supports our conclusion that retrieved values up to 3–5 pptv are below the detection limit of the actual dataset.

Nonetheless, our measurements impose constraints on the global distribution of upper tropospheric $NH_3$ concentrations which can be compared to results from model calculations. One of the first globally modeled distributions of $NH_3$ was presented by Dentener and Crutzen (1994, their

Fig. 2b). These calculations were based on the tropospheric transport model Moguntia with a horizontal resolution of $10° \times 10°$ with 10 layers up to 100 hPa in combination with, at that time, the first global emission inventory of $NH_3$ with the same resolution as the transport model. Yearly mean mixing ratios of below 2 pptv are modeled at upper tropospheric levels at mid- and high-latitudes. These are consistent with the MIPAS background values. However, at equatorial and sub-tropical

latitudes, annual mean values of some tens of pptv were simulated between 300 hPa ($\approx$9.5 km) and 200 hPa ($\approx$12.5 km) which are clearly larger than the MIPAS results. Dentener and Crutzen (1994) attributed these values to natural emissions in the tropics. The comparison with our results indicates that either these emissions might have been overestimated or the tropical sink processes of $NH_3$ underestimated. In their conclusions Dentener and Crutzen (1994) also mention high modeled con-

centrations of $NH_3$ in the free troposphere over India and China. However, since these enhancements were not quantified, they cannot be compared to our observations.

In contrast to the results of Dentener and Crutzen (1994), the zonal and yearly averages of modeled $NH_3$ shown in Feng and Penner (2007, Fig. 9) decrease to well below 10 pptv above 500 hPa ($\approx$6 km) also in tropical regions. Their aerosol chemistry transport model (Umich/IMPACT) had a

horizontal resolution of $2°$ latitude $\times 2.5°$ longitude with 26 layers up to 0.1 hPa using the $1° \times 1°$ global $NH_3$ emission inventory of Bouwman et al. (1997). Thus, these global model results are more compatible with the MIPAS dataset.

Globally resolved annual mean model results of $NH_3$ are given in Adams et al. (1999, Plate 3a). These data were based on runs with the general circulation model GISS GCM II-prime with

$4°$ latitude $\times 5°$ longitude horizontal resolution, nine vertical layers up to 10 hPa and $NH_3$ emissions according to Bouwman et al. (1997). Mean mixing ratios of about 3.2 pptv at the 200 hPa pressure level ($\approx$12.5/11 km in tropical/polar regions) are reported. At that pressure level, a slight gradient

between the two hemispheres is visible, with values of 0–1 pptv in the south and 3–10 pptv in the north. We do not recognize such a gradient in the background $NH_3$ concentrations from the MIPAS measurements, although, given our estimated error, we cannot conclusively refute such a gradient.

Regarding ammonium nitrate aerosol during the Asian monsoon season, Metzger et al. (2002) discuss their model results of enhanced values at upper tropospheric levels over Asia. They used the global chemistry-transport model TM3 with $7.5°$ latitude $\times$ $10°$ longitude horizontal resolution, 19 vertical levels and the EDGAR database for the emissions of $NH_3$. These high amounts of ammonium nitrate over Asia are attributed to in-situ production from $NH_3$ (and $HNO_3$) being convectively transported to upper tropospheric levels. The fact that $NH_3$ is not removed by dissolution in droplets and subsequent rainout is explained by the low acidity of convective clouds, such that only part of the $NH_3$ would be dissolved. Our observations support these results with respect to the enhanced amounts of $NH_3$ which obviously survive the uplift within the Asian monsoon circulation. This indicates that a part of the Asian tropopause aerosol layer (ATAL) (Vernier et al., 2011) might be composed of ammonium nitrate, ammonium sulfate or other ammonium containing particles.

Further, through a possible influence of the Asian monsoon on the composition of the tropical tropopause layer (TTL) by transport of ammonia or ammonium, our measurements may help to explain why in-situ measurements of aerosols in the TTL indicate that the sulfate appears to be mostly or fully neutralized (Froyd et al., 2009, 2010). Measurements of particle acidity hold potential to inform low $NH_3$ concentrations further in the background UT outside the Asian summer monsoon system.

## 6 Conclusions

We have presented first evidence of ammonia being present in the Earth's upper troposphere above 10 km by analysis of MIPAS infrared limb emission spectra. The region and period of detection is confined to the Asian summer monsoon system. Maximum average values of around 30 pptv over a three-month period have been retrieved, thus demonstrating that part of the $NH_3$ released at the ground survives the loss processes on its way to the upper troposphere. As suggested by Metzger et al. (2002), ammonia may form ammonium nitrate aerosols under those circumstances. Thus, our observations indicate a possible contribution of ammonium aerosols to the composition of the ATAL.

The detection of enhanced amounts of $NH_3$ in the western part of the Asian monsoon anticyclone during several years suggests that its lifetime is long enough to survive transport to areas far from the source region. The generally lower mixing ratios of $NH_3$ in the western compared to the eastern part indicate ongoing loss processes at high altitudes.

Unfortunately, in the literature there seem to exist no locally resolved model results of $NH_3$ during the monsoon period to which we could compare our observations. We anticipate that such simulations would be of value to improve our understanding of $NH_3$ loss processes and aerosol production

in the Asian monsoon. Also, airborne remote sensing observations, like the one planned within the EU project StratoClim with the GLORIA instrument on the Geophysica high flying aircraft, would strongly increase our knowledge about ammonia distributions in the Asian monsoon on a much finer time, horizontal and vertical resolution scale than the MIPAS dataset presented here.

Regarding the global distribution of upper tropospheric $NH_3$ outside the Asian monsoon, within this study we could provide upper limits in the range of a few $pptv$. This will help to constrain global models and to improve their results.

The $NH_3$ dataset is available upon request from the author or at http://www.imk-asf.kit.edu/english/308.php.

*Acknowledgements.* We thank Michelle Santee, a second reviewer and the editor Rolf Müller for their constructive comments and Bärbel Vogel for helpful discussions. Provision of MIPAS level-1b calibrated spectra by ESA and meteorological analysis data by ECMWF is acknowledged. The research leading to these results has received funding from the European Community's Seventh Framework Programme (FP7/2007-2013) under grant agreement 603557. R.V. is recipient of a KIT Distinguished Intl Scholar award, and acknowledges funding from the U.S. National Science Foundation EAGER award AGS-1452317. We acknowledge support by the Deutsche Forschungsgemeinschaft and Open Access Publishing Fund of the Karlsruhe Institute of Technology.

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

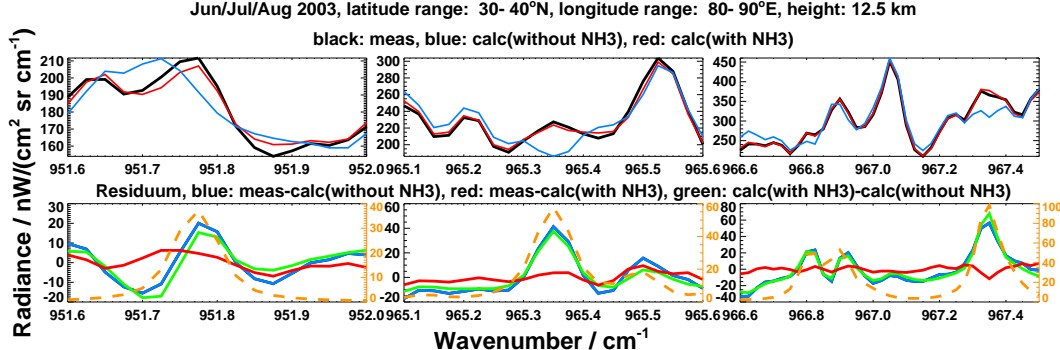

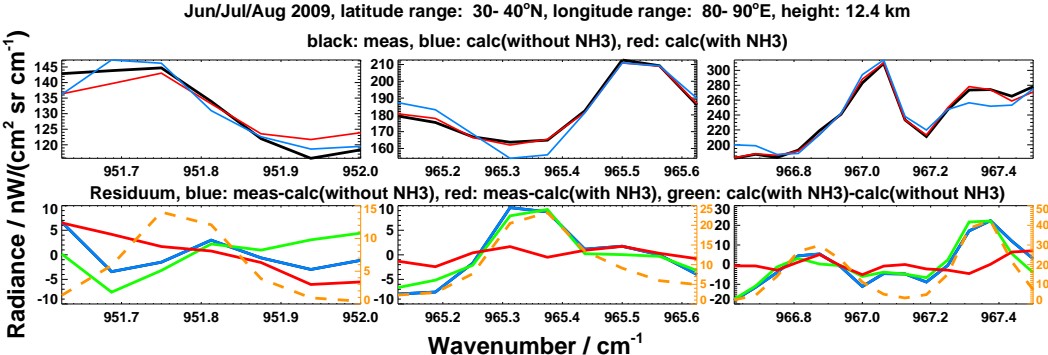

**Figure 1.** Spectral identification of $NH_3$ in MIPAS observations within the three spectral windows used for the retrieval (columns). The top two rows belong to the first observational period with higher spectral resolution. Rows 3 and 4 refer to the second period with lower spectral resolution. Rows 1 and 3 contain measured (black) and best fit spectra (blue: without, red: with $NH_3$). Row 2 and 4 show the spectral residuals without consideration of $NH_3$ (blue) and with $NH_3$ (red). Green curves in the second and fourth row represent the spectral features of $NH_3$ (calculation with $NH_3$ minus calculation without $NH_3$). To guide the eye, the orange dashed lines in rows 2 and 4 are simulated pure $NH_3$ spectra.

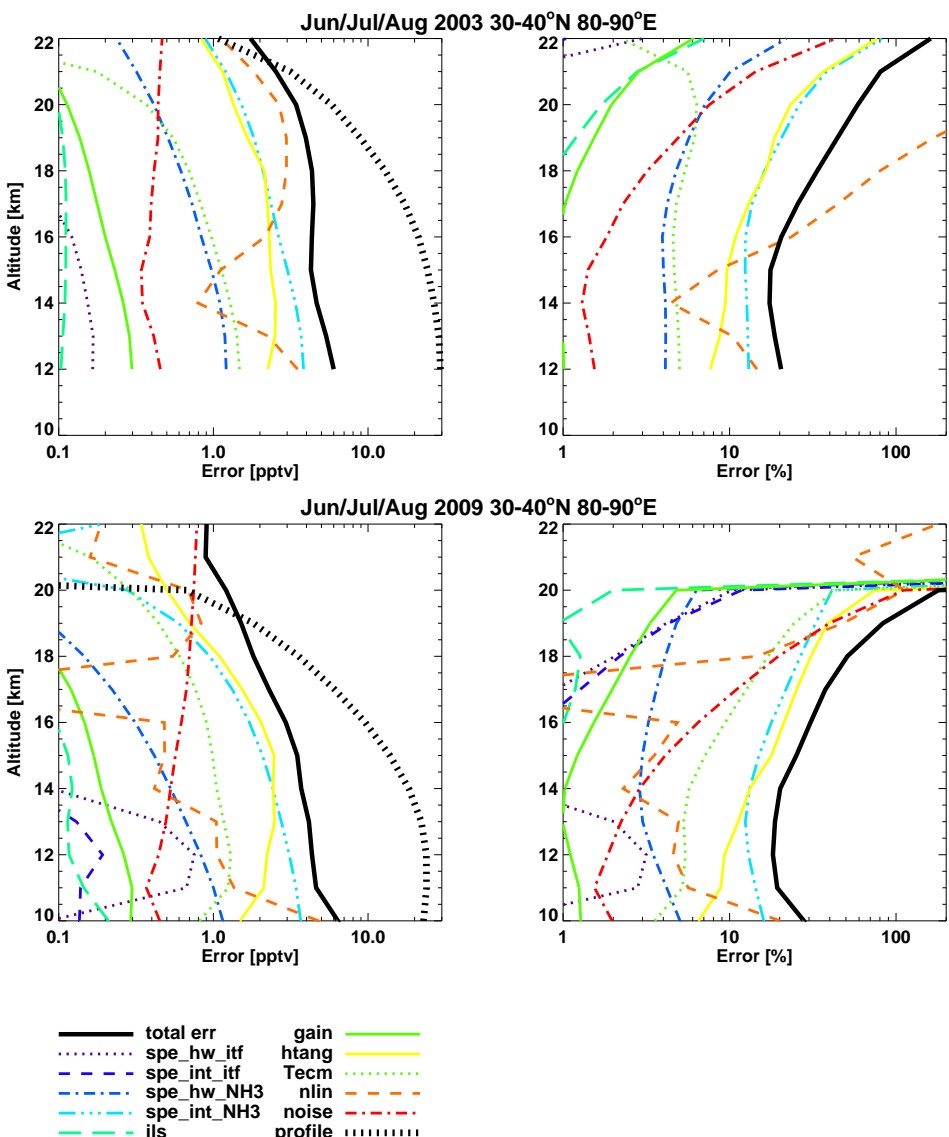

**Figure 2.** Estimated error profiles for two examples: one from measurement period 1 (June/July/August 2003, 30–40°N, 80–90°E) and one from period 2 (June/July/August 2009, 30–40°N, 80–90°E). The retrieved NH₃ profiles are shown as bold black dotted lines ("profile"). Abbreviations of the error sources are resolved in Tab. 1.

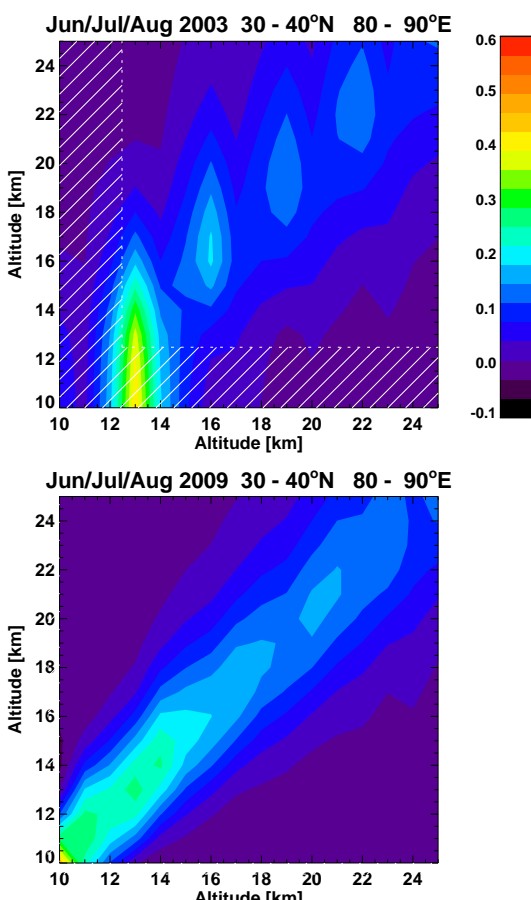

**Figure 3.** Averaging kernels of the MIPAS $NH_3$ retrieval during the first (top) and second (bottom) MIPAS measurement period. The number of degrees-of-freedom up to 25 km is 2.1 (top) and 3.5 (bottom). Hatched areas indicate altitudes below the lowest tangent height where no measurement information is available.

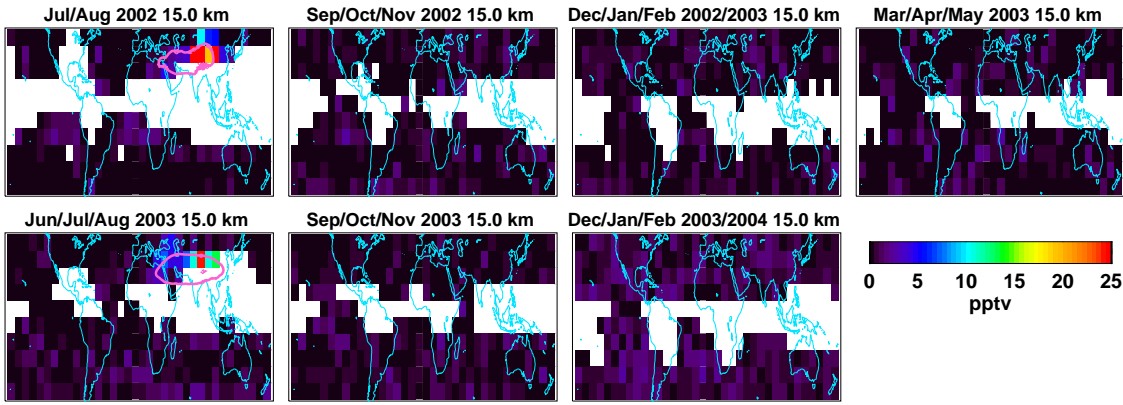

**Figure 4.** Distributions of $NH_3$ volume mixing ratios at 15 km altitude between $50°$N and $50°$S retrieved from MIPAS seasonal mean spectra during the first measurement period. Pixels where not enough spectra for averaging were available are left white. To guide the eye, the pink lines denote the approximate position of the Asian Monsoon Anticyclone. It is the $2 \times 10^{-6}$ $Km^2kg^{-1}s^{-1}$ contour of the mean potential vorticity for July/August in 2002 and June/July/August in 2003 at the potential temperature level of 370 K from the ECMWF ERA interim reanalysis (Dee et al., 2011) (e.g. Ploeger et al., 2015, and references therein).

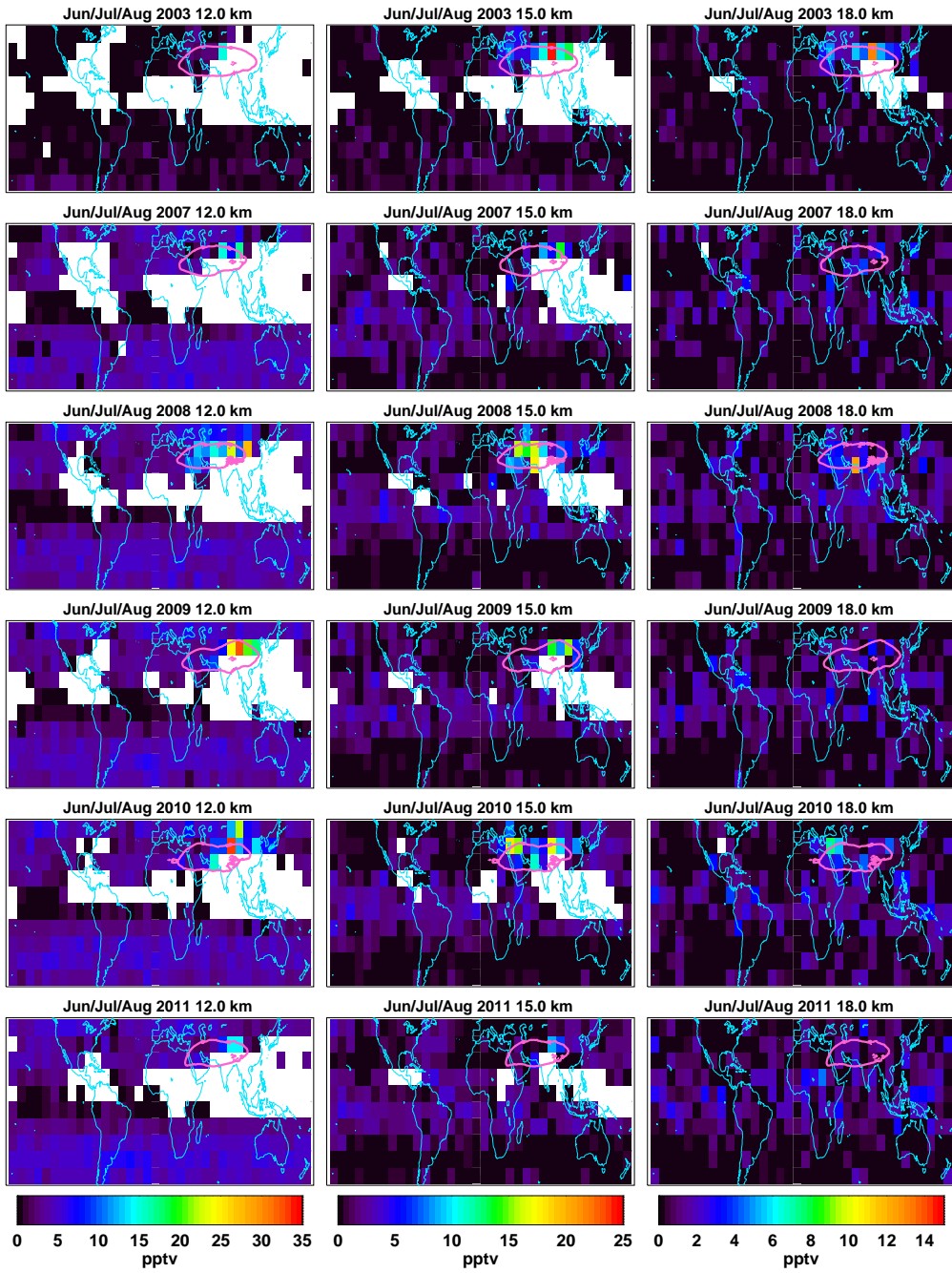

**Figure 5.** Distributions of $NH_3$ volume mixing ratios at 12 km, 15 km and 18 km altitude between $50°$N and $50°$S retrieved from MIPAS seasonal mean spectra during the Asian monsoon period for several years. Pixels where not enough spectra for averaging were available are left white. Pink contour lines denote the mean position of the Asian Monsoon Anticyclone for June/July/August as described in the caption of Fig. 4.

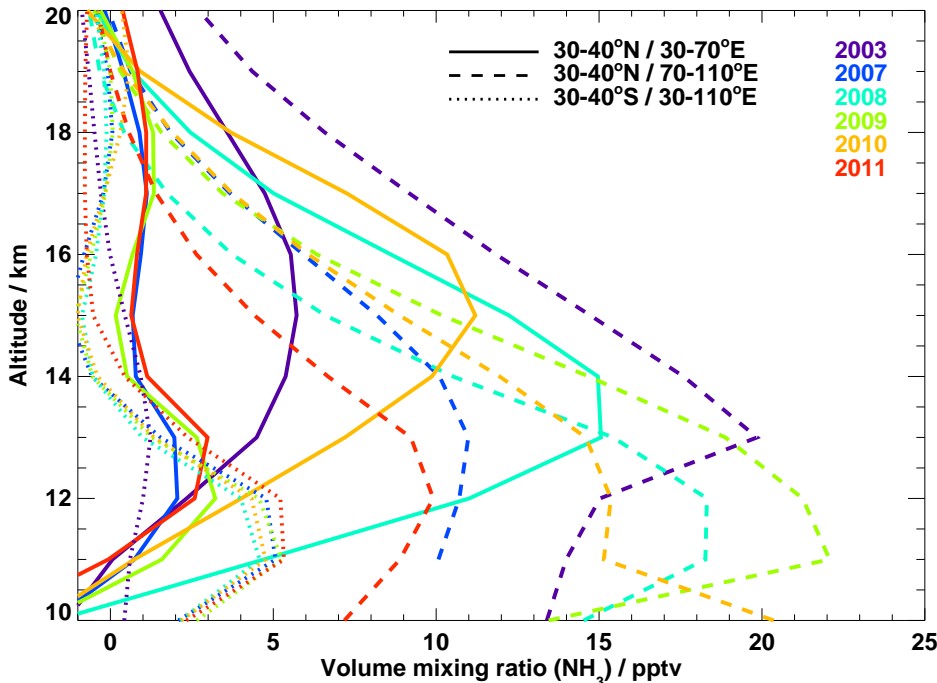

**Figure 6.** Mean profiles of $NH_3$ from MIPAS within the geographical range noted in the figure legend during June/July/August of each year. Solid: western part, dashed: eastern part of the Asian monsoon, dotted: reference profiles outside the monsoon in the southern hemisphere.

**Table 1.** Assumptions on uncertainties used for the error assessment in Fig. 2.

| Error source | Assumed uncertainty | Abbreviation |
|---|:---:|:---:|
| Spectral noise after apodization[1] | period 1: 2.2 (1.5–3.1) $\mathrm{nW/(cm^2\,sr\,cm^{-1})}$ | noise |
| | period 2: 1.3 (0.8–1.8) $\mathrm{nW/(cm^2\,sr\,cm^{-1})}$ | |
| Instrument line shape[2] | 3% | ils |
| Instrument gain calibration[3] | 1% | gain |
| Tangent altitude[4] | 300 m | htang |
| Temperature[5] | 2 K below/5 K above 35 km | Tecm |
| Retrieval from averaged spectra[6] | | nlin |
| Air-broadened half-width of $NH_3$ lines[7] | 10% | spe_hw_NH$_3$ |
| Intensity of $NH_3$ lines[7] | 15% | spe_int_NH$_3$ |
| Air-broadened half-width of interfering gas lines[7] | 15% | spe_hw_itf |
| Intensity of interfering gas lines[7] | 5% | spe_int_itf |

[1] ESA l1b dataset, depending on number of co-added spectra; [2] F. Hase, pers. comm., Höpfner et al. (2007); [3] Kleinert et al. (2007); [4] von Clarmann et al. (2003); von Clarmann et al. (2009); Kiefer et al. (2007); [5] ECMWF uncertainty Höpfner et al. (2013); [6] Höpfner et al. (2009); Höpfner et al. (2013); [7] HITRAN 2012 spectral line errors Rothman et al. (2013)