# Peer review of "First detection of ammonia (NH3) in the Asian summer monsoon upper troposphere"

_Atmospheric Chemistry and Physics, 2016_

## Referee Comment (RC1) · M. Santee (Referee) · 30 Jun 2016

This manuscript presents the first unequivocal detection of enhanced NH3 in the UTLS. Retrievals of NH3 from seasonally averaged MIPAS spectra in 10 deg x 10 deg bins are described and their errors estimated through systematic uncertainty analysis. Enhanced UTLS NH3 is found only within the Asian summer monsoon anticyclone. The implications of these enhancements for the ATAL are discussed.

I am not an expert on the sources and sinks controlling atmospheric NH3, so I cannot critically evaluate much of the background information presented here, but the authors appear to have done a very diligent job of documenting the previous literature and placing their new measurements into context. In general I think that the analysis is sound, and the results are well presented. I have only a few very minor substantive

comments for the authors to consider, all of which should be quite easy to address. In addition, because the manuscript is generally well written, I have made the effort to correct a few typos, and I also added a number of suggestions for other small wording changes that I feel would further enhance the quality of the paper – in truth, I would not have bothered making most of these suggestions had the manuscript not been so polished already!

Specific substantive comments:

– Just to avoid any potential for ambiguity (since there is also an Asian winter monsoon), I suggest that the word "summer" be inserted before "monsoon" in a few more places in the manuscript, for example: the title of the article, the Abstract (L3), the Discussion section (L193), and the Conclusions (L243).

– L142-143: Shouldn't the total error be the RSS of the individual sources of uncertainty? That is, shouldn't the error components being summed be squared?

– Figure 4 shows the seasonal distributions of NH3 during MIPAS period 1. But I am not sure that it is necessary to show all 7 seasons in that interval, especially given that the first panel covers only July and August 2002 and is thus not completely comparable to the 3-month averages depicted in the other panels. Perhaps the information could be conveyed with just one row of 4 maps, starting with MAM 2003, then JJA 2003, SON 2003, and ending with DJF 2003/2004. Then the fact that the other seasons from period 1 show similar results could simply be stated in words. For completeness, such a statement about the other seasons in period 2 should be made in any case, as should a statement about other altitudes in period 1.

– When I first read through Section 4, I thought that although there may not be any correlative measurements of UTLS NH3 to validate the MIPAS retrievals against, there surely must be some model simulations that could provide a zeroth-order "sanity check" on the morphology if not the magnitude of the retrieved distribution. It turns out that model results (or the lack thereof) are discussed at length in Section 5, but it might

be useful to add a sentence in this section that points forward to that discussion, so that readers do not assume at this point that opportunities for validation have been overlooked.

– It is stated (L177-178) that: "the maximum concentrations of NH3 are always larger within the eastern part of the Asian monsoon". However, this statement is only true at certain altitudes; it is not the case above 13 km in 2008 or above 15 km in 2010.

– It is noted (L181) that in the western portion of the monsoon region enhanced NH3 "can only be observed during the years 2003, 2008, and 2010". As written, this makes such enhancements sound like a rare occurrence. But that sample includes half of the years observed.

– L196-197: Nor could random errors account for the enhancements appearing only in one season.

– L250-254: The point about the differences in the altitudes of the peaks in the NH3 profiles in the eastern and western parts of the monsoon region being consistent with the "general view" has not been made previously in the manuscript, and it seems to me that it would be more appropriate to make such a point for the first time in the Discussion section (or in Section 4 where the differences in the two regions are initially discussed) and not the Conclusions. Moreover, a reference or two should be provided for the description of the "general view" of the monsoon system.

– Fig 1: The orange lines are helpful but somewhat hard to see. It might be better to use solid or dashed rather than dotted lines.

Typos and other minor wording and grammar corrections / suggestions:

– L8: "aersol"

– L14: "bulk" would be better than "wealth"

– L15: "by use" –> "the use"

Interactive
comment

– L16: delete comma after "and"

– L20: add a comma after "(NH4)2SO4"

– L24-25: "also cirrus clouds might be affected" –> "cirrus clouds might also be affected"

– L27: "respect of" –> "respect to"

– L33: "prospects for"

– L39: add a comma after "Beer et al. (2008)"

– L49: "the ground"

– L53: "vast" is not quite the right word. I suggest either replacing it with "severe" or simply deleting it.

– The paragraph in L55-58 is all one sentence, and it is followed by another short paragraph in L59-61. It seems to me that these two short paragraphs could be combined into one.

– L61: I found the last part of this sentence confusing and had to read it twice to understand the meaning. I suggest rewording as: "... restricted NH3 concentrations to the sub-pptv range at altitudes between 8 and 10 km".

– L62: "In the case". Also add a comma after "instruments"

– L66: add a comma after "NH3"

– L67: "like" –> "such as"

– L74 & 76: add commas after "(period 1)" and "(period 2)"

– L75: "UTLS" should be spelled out the first time it is used; also "in the case"

– L78: "in the horizontal"
– L83: "on the basis"

– L86: "those investigations" would be clearer

– L87: add a comma after "intervals"

– L89-90: (1) "the meridional", (2) add a comma after "direction", (3) "To obtain at least a reduction of the spectral noise of at least" –> "To reduce the spectral noise by at least"

– L93: "As cloud" –> "For the cloud"

– L96: where –> "whereby"

– L102: "oder"

– L106: add a comma after "970 cm-1"

– L108: "simultaneously to" –> "simultaneously with"

– L110: delete "subsequent"

– L117: add a comma after "retrieval"

– L118: delete "two"

– L123: "both" –> "the two"

– L125: "like" –> "such as"

– L128: it would be good to add "(orange curves)" after "ammonia lines"

– L128-129: move "are" from before "fitted" to after "account" and delete the comma there

– L144: add ", right panels" after "70-80%"

– L154: add a comma after "20 km"

[Figure]

– L173: "parts"

– L174-175: "curves show the NH3 mean profiles for all years"

– L178: "vmr" is not used elsewhere in the text, and I don't think it should be used here either.

– L183: "always located"

– L186: add a comma after "5 pptv"

– L187: "in the case"; also "indicated by"

– L198: delete "the" before "15 ppbv"

– L202: "which amounts up to" –> "that reaches"

– L203: "both" –> "the two"

– L208: "has been" –> "was"

– L210: "in good agreement with" may be too strong in this case (given the uncertainty in the NH3 data); "consistent with" may be more appropriate

– L214: "overestimated" would be better than "over-"

– L218: "In contrast to the results of"

– L223: "both" –> "the two"; also add a comma after "visible"

– L224: "albeit" –> "although"; also "given" would be better than "compared to" and "conclusively" would be better than "clearly"

– L231: "clouds by which" –> "clouds, such that"

– L237: move the comma after "(TTL)" to after "ammonium"

– L241: "first evidence for" –> "the first evidence of"

– L243: "three-monthly" –> "three-month"

– L244: delete the comma after "thus"; also "at the ground"

– L248-254: No need to capitalize "Western" and "Eastern", or "West" and "East"

– L249: "transport to areas far from"

– L254: "ongoing"

– L258-259: (1) "ones", (2) either delete the comma after "aircraft" or add one after "observations" (i.e., the part of the sentence from "like" to "aircraft" should be set off by two commas or none, not one); (3) I suggest saying "would" rather than "will"

– L260: delete the hyphen after "time"

– Fig 2 caption: "vmr" is not used elsewhere in the figure captions, and I don't think it should be used here either

– Fig 5 caption: "seasonal mean spectra for several years during the Asian monsoon period" would be better as "seasonal mean spectra during the Asian monsoon period for several years"

– Fig 6 caption: "westerly" –> "western"; "easterly" –> "eastern"

---

## Referee Comment (RC2) · Anonymous Referee #2 · 30 Sep 2016

In this paper, the authors present the retrieval of NH3 from MIPAS/Envisat spectra using the averaging technique that has been used previously for BrONO2 and SO2. They have detected enhanced amounts of NH3 (e.g. above the MIPAS detection limit) within the Asian monsoon on a three month average basis (June, July and August). These are the first upper troposphere measurement of NH3 in the Asian monsoon. These results are discussed with respect to other measurements and model studies.

This is a useful and interesting contribution. I believe that it will be suitable for publication in ACP once the following comments have been addressed.

General comments:

In the introduction, I found that the discussion of previous measurements needs to be clarified and enhanced. The description of what has been done via in-situ and remote

sensing techniques should be made much more distinct. These were somewhat mixed together in the text. Specifically, the platforms used (e.g. ground, balloon, satellite) and the altitude range for measurement sensitivity needs to be described more thoroughly. In particular, the authors should establish how far into the UT the nadir and balloon measurements reach to support their "first" measurement claim within the Asian monsoon.

Because different coordinate systems are used by the different measurements, when discussing altitude or pressure ranges for current or previous results, both z and p should be given to help the reader to make these connections clearly.

In the discussion of the definition of the detection limit, it would be useful to give further description of the impact of choosing 2-sigma versus 1-sigma as the limit. How frequently are the enhancements above 15 pptv (3-sigma)? Would this choice of limit impact detection in certain years or all years?

When discussing the model results, I missed a bit more detail on the type of model results used. This could be added to the introduction or put into the discussion section. Are there any dependences of results on the met. fields driving the models or any emissions included? In the conclusion, the term "locally resolved model" was used. This stated lack of model results needs to be supported better in the discussion

Specific comments:

L24-27 Could the authors clarify a bit further how this would impact clouds? Increase their presence?

L65-68 While the quote "tentative identification" was taken from Coheur et al., a profile was retrieved from the ACE measurements. This should be clarified in the introduction.

L89-91 Please provide the average number of spectra used in the monthly averages. Does this vary significantly by year (maybe based on cloud presence?) Also, what is the typical signal to noise ratio of the averaged spectra? Is it much better than the

minimum stated? Are the spectra evenly distributed throughout each month?

L112-116 It seems that the spectral windows between the two periods differ by one spectral "grid point". Could the authors comment on why this seemly small change was necessary?

L130-132 If MWs 2 and 3 show the "best fit", why is MW 1 included? Can it be omitted?

L154-155 It would be useful to state the vertical resolution also for the altitude levels used later on in the discussion. Is it closer to the higher or lower value?

L172-179 Are the same months used for the background as the Asian Monsoon average? To identify the grid boxes within the Asian monsoon, is the ERA Interim contour used or a fixed latitude-longitude box? If a box, does this change by year or month?

L241-242 This sentence seems to be overstating the results as the authors report that UT NH3 measurements over Germany were made by Ziereis and Arnold. This should be clarified by the authors and supported by the text.

Data availability should be discussed in the paper at end of conclusions or in a separate data section.

Technical comments:

L8 aersol should be aerosol.

L60 Are these transmission or emission spectra?

L75 Has UTLS been defined prior to this in the text? Also, the altitude range used for the UTLS should be specified in the text.

L78 Should be "in the horizontal" direction.

L102 "oder" should be "order".

L108 simultaneously with NH3.

L125 The pure NH3 spectrum line should be mentioned here. Also, how it was calculated. Retrieved VMR?

L178&180-181 Is vmr defined? It does not seem to be used consistently throughout. This should be fixed.

L214 Do you mean over-estimated here?

Unused references for Kiefer et al., Kleinert et al., Hoepfner et al., 2007, Ploeger et al., von Clarmann et al.

Figures 4 and 5 The latitude and longitude markers should be included on the plots to make it clearer.

Table 1 There seems to be an assumed uncertainty missing for one line.

---

## Author Comment (AC1) · 18 Oct 2016

We would like to thank Michelle Santee (referee 1) for her valuable comments and corrections all of them leading to an improvement of our manuscript. Comments and questions of the referee are marked in bold face and manuscript changes in italics.

**Just to avoid any potential for ambiguity (since there is also an Asian winter monsoon), I suggest that the word "summer" be inserted before "monsoon" in a few more places in the manuscript, for example: the title of the article, the Abstract (L3), the Discussion section (L193), and the Conclusions (L243).**

We agree and have inserted *"summer"* at the suggested locations.

**L142-143: Shouldn't the total error be the RSS of the individual sources of un-**

[Figure]

**certainty? That is, shouldn't the error components being summed be squared?**

Right, this is an error in the text (not in the Figure). We have corrected the text accordingly by inserting *"squared"* before *"error components"*.

**Figure 4 shows the seasonal distributions of NH$_3$ during MIPAS period 1. But I am not sure that it is necessary to show all 7 seasons in that interval, especially given that the first panel covers only July and August 2002 and is thus not completely comparable to the 3-month averages depicted in the other panels. Perhaps the information could be conveyed with just one row of 4 maps, starting with MAM 2003, then JJA 2003, SON 2003, and ending with DJF 2003/2004. Then the fact that the other seasons from period 1 show similar results could simply be stated in words. For completeness, such a statement about the other seasons in period 2 should be made in any case, as should a statement about other altitudes in period 1.**

The text has been changed according to the reviewer's suggestions:

From: *"During all other seasons and outside the region influenced by the Asian monsoon, no similarly high concentrations of NH$_3$ can be found."*

To: *"During all other seasons of the two MIPAS periods and outside the region influenced by the Asian monsoon, no similarly high concentrations of NH$_3$ can be found within the entire altitude region covered by our measurements."*

Regarding the proposed update of Figure 4, we tend not to change it in order, (a) to demonstrate that also in Jul/Aug 2002 there have been enhancements of NH$_3$ in the Asian monsoon region, and, (b) to cover one period of the MIPAS observations entirely.

**When I first read through Section 4, I thought that although there may not be any correlative measurements of UTLS NH$_3$ to validate the MIPAS retrievals against, there surely must be some model simulations that could provide a zeroth-order "sanity check" on the morphology if not the magnitude of the retrieved distribu-**

**tion. It turns out that model results (or the lack thereof) are discussed at length in Section 5, but it might be useful to add a sentence in this section that points forward to that discussion, so that readers do not assume at this point that opportunities for validation have been overlooked.**

We agree and have added some text at the end of Section 4:

*"Due to the lack of ammonia observations in the upper troposphere, we cannot validate our dataset with correlative measurements. However, in the next section we discuss its plausibility by comparing with the few previous observations and atmospheric model results."*

**It is stated (L177–178) that: "the maximum concentrations of $NH_3$ are always larger within the eastern part of the Asian monsoon". However, this statement is only true at certain altitudes; it is not the case above 13 km in 2008 or above 15 km in 2010.**

This wording might be misleading. What we meant here relates to the maximum values over the whole altitude range of the profile. We have tried to make it clearer by changing the sentence to:

*"The profiles in the region of the Asian monsoon reveal that the maximum concentrations over the whole altitude range within one year are always larger within the eastern part of the Asian monsoon compared to the western part. Maximum concentrations of $NH_3$ in the eastern part reach about 10–22 pptv at 11–13 km altitude."*

**It is noted (L181) that in the western portion of the monsoon region enhanced $NH_3$ "can only be observed during the years 2003, 2008, and 2010". As written, this makes such enhancements sound like a rare occurrence. But that sample includes half of the years observed.**

Agreed: we have skipped the word *"only"*.

**L196–197: Nor could random errors account for the enhancements appearing**

**only in one season.**

Thanks for pointing to this argument. We have added it by changing the sentence to: *"However, random errors cannot explain why the enhancements should appear in a contiguous geographical pattern nor could they account for the enhancements appearing only in one season."*

**L250–254: The point about the differences in the altitudes of the peaks in the NH$_3$ profiles in the eastern and western parts of the monsoon region being consistent with the "general view" has not been made previously in the manuscript, and it seems to me that it would be more appropriate to make such a point for the first time in the Discussion section (or in Section 4 where the differences in the two regions are initially discussed) and not the Conclusions. Moreover, a reference or two should be provided for the description of the "general view" of the monsoon system.**

In the revised version we have (1) moved this part to Section 4, as suggested by the referee, and, (2) weakened the statement on the "general view", since we could not support it clearly on basis of published material.

Text added after line 184 of the original manuscript:

*"The position of the NH$_3$ maximum at higher altitudes in the western compared to the eastern part of the monsoon system might be due to convective uplift of boundary layer air in the east followed by upper tropospheric transport and further uplift towards the west. Such an uplift of air from east to west is indicated in Vogel et al. (2014, Fig. 10) by trajectory calculations, however mainly located at the border of the anticyclone.*

**Fig 1: The orange lines are helpful but somewhat hard to see. It might be better to use solid or dashed rather than dotted lines.**

OK, we have changed them to dashed style and used a thicker line width.

**Typos and other minor wording and grammar corrections / suggestions:**

Thanks for the list of technical corrections! We have implemented all of them as suggested.

**New references**

Vogel, B., Günther, G., Müller, R., Grooß, J.-U., Hoor, P., Krämer, M., Müller, S., Zahn, A., and Riese, M.: Fast transport from Southeast Asia boundary layer sources to northern Europe: rapid uplift in typhoons and eastward eddy shedding of the Asian monsoon anticyclone, Atmos. Chem. Phys., 14, 12 745–12 762, 600 doi:10.5194/acp-14-12745-2014, http://www.atmos-chem-phys.net/14/12745/2014/, 2014.

---

## Author Comment (AC2) · 18 Oct 2016

We thank referee 2 for carefully evaluating our manuscript. For the revised version we have tried to take all suggestions into account. Comments and questions of the referee are marked in bold face and manuscript changes in italics.

**In the introduction, I found that the discussion of previous measurements needs to be clarified and enhanced. The description of what has been done via in-situ and remote sensing techniques should be made much more distinct. These were somewhat mixed together in the text. Specifically, the platforms used (e.g. ground, balloon, satellite) and the altitude range for measurement sensitivity needs to be described more thoroughly. In particular, the authors should establish how far into the UT the nadir and balloon measurements reach to support**

[Figure]

**their "first" measurement claim within the Asian monsoon.**

We agree with the referee that this part of the introduction was difficult to read and a bit too compressed. Thus, we have restructured the description of $NH_3$ observations to clearly separate in-situ from remote sensing observations. Further we have added more information about the platforms and the altitude regions of sensitivity to support our claim of "first evidence for $NH_3$". Lines 37–68 of the paper will be replaced by the following text in the revised version:

[revised manuscript text omitted]

**Because different coordinate systems are used by the different measurements, when discussing altitude or pressure ranges for current or previous results, both z and p should be given to help the reader to make these connections clearly.**

According to the reviewers suggestion we have indicated the approximate altitudes everywhere in the manuscript where only pressure was given.

Text changes:

Line 260: *"between 300 hPa ($\approx$9.5 km) and 200 hPa ($\approx$12.5 km)"*

Line 268: *"above 500 hPa ($\approx$6 km) also in tropical regions"*

Line 271: *"at the 200 hPa pressure level ($\approx$12.5/11 km in tropical/polar regions)"*

**In the discussion of the definition of the detection limit, it would be useful to give further description of the impact of choosing 2-sigma versus 1-sigma as the limit. How frequently are the enhancements above 15 pptv (3-sigma)? Would this choice of limit impact detection in certain years or all years?**

We have added this information in the updated version of the manuscript.

Lines 196–198 have been extended as follows:

*"For example within the data shown in Fig. 5 at 12 km, there are 176 values larger than 5 pptv outside the region 20–50$^\circ$ N $\times$ 30–120$^\circ$ E compared to 55 inside. However, at the 15 km level, only 5 data points exceed 5 pptv outside but 37 inside. Using 2-$\sigma$, there are no data points outside compared to 23 and 15 exceeding 10 pptv inside the region at 12 km and 15 km, respectively. Further, the detected enhancements inside the monsoon region are in many cases (13 times at 12 km and 8 times at 15 km) even*

*above 15 pptv and, thus, larger than a 3-σ limit. Temporally resolved, values above 10 pptv in the monsoon region exist during all years at both altitude levels with the exception of 2011 at 15 km. 15 pptv are exceeded at 10 km in 2003–2010 and at 15 km in all years but 2007 and 2011."*

**When discussing the model results, I missed a bit more detail on the type of model results used. This could be added to the introduction or put into the discussion section. Are there any dependences of results on the met. fields driving the models or any emissions included? In the conclusion, the term "locally resolved model" was used. This stated lack of model results needs to be supported better in the discussion.**

As suggested by the referee, we have added more information on the models used. The four quoted publications use e.g. different dynamic kernels and emission inventories. However, it is outside the scope of this paper to explain the reasons for different model results regarding the global distribution of ammonia. Further, in the discussion, we state a lack of published model data resolving $NH_3$ within the Asian monsoon period. Thus, we do not exclude the existence of model runs from which this information could be drawn - we just have not found those in literature.

Text changes:

Line 209: *"These calculations were based on the tropospheric transport model Moguntia with a horizontal resolution of 10° × 10° with 10 layers up to 100 hPa in combination with, at that time, the first global emission inventory of $NH_3$ with the same resolution as the transport model."*

Line 219: *"Their aerosol chemistry transport model (Umich/IMPACT) had a horizontal resolution of 2° latitude × 2.5° longitude with 26 layers up to 0.1 hPa using the 1° × 1° global $NH_3$ emission inventory of Bouwman et al., 1997."*

Line 221: *"These data were based on runs with the general circulation model GISS*

*GCM II-prime with 4° latitude × 5° longitude horizontal resolution, nine vertical layers up to 10 hPa and NH$_3$ emissions according to Bouwman et al., 1997."*

Line 228: *"They used the global chemistry-transport model TM3 with 7.5° latitude × 10° longitude horizontal resolution, 19 vertical levels and the EDGAR database for the emissions of NH$_3$."*

**Specific comments:**

**L24-27 Could the authors clarify a bit further how this would impact clouds? Increase their presence?**

Thanks for this comment. In response, we have included, the paper by Abbatt et al. (2006) who investigated the effect of ammonium sulfate on heterogeneous cirrus nucleation resulting in fewer and larger ice particles compared to the case of homogeneous nucleation.

Text changes line 27ff:

*"Such a heterogeneous nucleation pathway might influence size and number of cirrus particles and, consequently their radiative impact (Abbatt et al., 2006)."*

**L65-68 While the quote "tentative identification" was taken from Coheur et al., a profile was retrieved from the ACE measurements. This should be clarified in the introduction.**

Even if there was no clear visible "evidence" for ammonia in the spectra, a retrieval can be performed. We have added this information in the revised manuscript.

Added text: *"Nonetheless, Coheur et al., 2007 derived a vertical profile of NH$_3$ between 6.5 and 17 km with typical values of less than 20 pptv and a maximum of about 50 pptv at 8 km."*

**L89-91 Please provide the average number of spectra used in the monthly averages. Does this vary significantly by year (maybe based on cloud presence?)**
**Interactive comment**

**Also, what is the typical signal to noise ratio of the averaged spectra? Is it much better than the minimum stated? Are the spectra evenly distributed throughout each month?**

According the referee's suggestion, we have added two figures in the supplement showing the number of co-added spectra per data bin for each sub-plot of Figures 4 and 5 of the paper. The mean number of co-added spectra is by a factor of 2-3 higher than the minimum value of 25, and, thus the noise is reduced by factors of about 0.5-0.6 compared to the noise reduction obtained by the minimum allowed number of co-added spectra. The temporal variation of this number between years (see supplement, Fig. 2) is mainly determined (a) by the different sampling frequencies between phase 1 and phase 2 of the MIPAS operational period and the reduced sampling in 2007 compared to the later years. Its variation between different months is not very strong (see supplement, Fig. 1).

Added text at line 91:

*"The resulting mean number of co-added spectra per time/latitude/longitude bin varies from 53–56 for the years 2003 and 2007 to 65–70 for 2008–2011 reaching maximum numbers of around 140 (see supplemental material for the detailed geographical and temporal distribution). This leads to a typical reduction of the spectral noise by 0.1 ranging from 0.2 to 0.08 and signal-to-noise values resulting in retrieval errors near and below 1 pptv of $NH_3$ (see detailed error estimation below)."*

**L112-116 It seems that the spectral windows between the two periods differ by one spectral "grid point". Could the authors comment on why this seemly small change was necessary?**

These changes have been necessary for technical reasons due to the change in the spectral grid from MIPAS phase 1 (0.025 $cm^{-1}$) to phase 2 (0.0625 $cm^{-1}$).

**L130-132 If MWs 2 and 3 show the "best fit", why is MW 1 included? Can it be**

[Figure]

**omitted?**

In case of the high spectral resolution, a clear improvement is visible also in MW 1. This spectral signal is also fitted in MW 1 in case of the lower resolution, reducing e.g. the peak in the residual around 951.8 cm$^{-1}$ and contributing information. However, due to the lower strength of the NH$_3$ line in MW1, this is much less obvious. We agree that it should be possible to skip MW 1 entirely from the retrieval. However, we have not taken this option in order to keep the differences between the retrievals between MIPAS phase 1 and phase 2 as small as possible.

**L154-155 It would be useful to state the vertical resolution also for the altitude levels used later on in the discussion. Is it closer to the higher or lower value?**

We have specified the vertical resolutions more clearly in the revised manuscript.

Changed text at lines 154-155:

*"The globally average vertical resolution at the altitude levels 12, 15, and 18 km, which are discussed in more detail below, is 6.6 km, 7.9 km and 8.8 km during period 1 and 3.5 km, 4.3 km, and 5.6 km during period 2."*

**L172-179 Are the same months used for the background as the Asian Monsoon average? To identify the grid boxes within the Asian monsoon, is the ERA Interim contour used or a fixed latitude-longitude box? If a box, does this change by year or month?**

Yes, the same months are used for the background values. We have specified this better in the revised version. Regarding the grid boxes: a temporally fixed box area has been applied as stated in the text (lines 172-174) and in the legend within Figure 6.

Changed text 174–176:

*"In the same Figure, the dotted curves show the NH$_3$ mean Jun/Jul/Aug profiles for*

*all years outside the Asian monsoon area, for the same longitude and latitude range (30–110° E, 30–40° S) of the southern hemisphere."*

**L241-242 This sentence seems to be overstating the results as the authors report that UT NH$_3$ measurements over Germany were made by Ziereis and Arnold. This should be clarified by the authors and supported by the text.**

At their highest levels, 9 and 10 km, Ziereis and Arnold present upper limits for NH$_3$, i.e. not a detection. We have clarified this in the revised version:

Changed text lines 241–242:

*"We have presented first evidence of ammonia being present in the Earth's upper troposphere above 10 km by analysis of MIPAS infrared limb emission spectra."*

**Data availability should be discussed in the paper at end of conclusions or in a separate data section.**

Added text at the end of the conclusions:

*"The NH$_3$ dataset is available upon request from the author or at http://www.imk-asf.kit.edu/english/308.php."*

**Technical comments:**

**L8 aersol should be aerosol.**

Corrected.

**L60 Are these transmission or emission spectra?**

This has been specified more clearly in the revised version:

*"balloon-borne limb solar absorption spectra"*

**L75 Has UTLS been defined prior to this in the text? Also, the altitude range used for the UTLS should be specified in the text.**

We have deleted "UTLS" in the new version since it appeared only three times where it could be replaced by more quantitative expressions:

Line 75: *"in the UTLS"* replaced by *"up to about 42 km"*

Line 78: *"27 tangent levels with 1.5 km steps in the UTLS"* replaced by *"27 tangent levels up to about 70 km altitude with 1.5 km steps up to ≈23 km"*

Line 99: *"within the region of the UTLS"* has been deleted, since the entire altitude grid has a spacing of 1 km.

**L78 Should be "in the horizontal" direction.**

Corrected.

**L102 "oder" should be "order".**

Corrected.

**L108 simultaneously with NH$_3$.**

Corrected.

**New references**

Abbatt, J. P. D., Benz, S., Cziczo, D. J., Kanji, Z., Lohmann, U., and Möhler, O.: Solid Ammonium Sulfate Aerosols as Ice Nuclei: A Pathway for Cirrus Cloud Formation, Science, doi:10.1126/science.1129726, 2006.

Schiferl, L. D., Heald, C. L., Van Damme, M., Clarisse, L., Clerbaux, C., Coheur, P.-F., Nowak, J. B., Neuman, J. A., Herndon, S. C., Roscioli, J. R., and Eilerman, S. J.: Interannual variability of ammonia concentrations over the United States: sources and implications, Atmos. Chem. Phys., 16, 12 305–12 328, doi:10.5194/acp-16-12305-2016, 2016.

Please also note the supplement to this comment:

[Figure]

http://www.atmos-chem-phys-discuss.net/acp-2016-392/acp-2016-392-AC2-supplement.pdf

[Figure]

**Supplement:**

[Figure]

Figure 1: Same as Fig. 4 of the paper, but with a color scale indicating the number of observations used for calculating the averaged spectra.

[Figure]

Figure 2: Same as Fig. 5 of the paper, but with a color scale indicating the number of observations used for calculating the averaged spectra..